# Flexible Phase Change Material Fiber: A Simple Route to Thermal Energy Control Textiles

**DOI:** 10.3390/ma14020401

**Published:** 2021-01-15

**Authors:** Yurong Yan, Weipei Li, Ruitian Zhu, Chao Lin, Rudolf Hufenus

**Affiliations:** 1School of Materials Science and Engineering, South China University of Technology, Guangzhou 510640, China; liweipeidc@163.com (W.L.); linchaocm@163.com (C.L.); 2Key Lab Guangdong High Property & Functional Polymer Materials, Guangzhou 510640, China; 3Laboratory for Advanced Fibers, Empa, Swiss Federal Laboratories for Materials Science and Technology, Lerchenfeldstrasse 5, 9014 St. Gallen, Switzerland; 4Guangzhou Inspection Testing and Certification Group Co., Ltd., Guangzhou 511447, China; zhurt@gttc.net.cn

**Keywords:** phase change material, polyethylene glycol, fluid-filled fiber, thermal energy control

## Abstract

A flexible hollow polypropylene (PP) fiber was filled with the phase change material (PCM) polyethylene glycol 1000 (PEG1000), using a micro-fluidic filling technology. The fiber’s latent heat storage and release, thermal reversibility, mechanical properties, and phase change behavior as a function of fiber drawing, were characterized. Differential scanning calorimetry (DSC) results showed that both enthalpies of melting and solidification of the PCM encased within the PP fiber were scarcely influenced by the constraint, compared to unconfined PEG1000. The maximum filling ratio of PEG1000 within the tubular PP filament was ~83 wt.%, and the encapsulation efficiencies and heat loss percentages were 96.7% and 7.65% for as-spun fibers and 93.7% and 1.53% for post-drawn fibers, respectively. Weak adherence of PEG on the inner surface of the PP fibers favored bubble formation and aggregating at the core–sheath interface, which led to different crystallization behavior of PEG1000 at the interface and in the PCM matrix. The thermal stability of PEG was unaffected by the PP encasing; only the decomposition temperature, corresponding to 50% weight loss of PEG1000 inside the PP fiber, was a little higher compared to that of pure PEG1000. Cycling heating and cooling tests proved the reversibility of latent heat release and storage properties, and the reliability of the PCM fiber.

## 1. Introduction 

Clothing systems with adaptive thermoregulation and acclimatization properties are attracting a lot of attention [1]. Phase change materials (PCMs) offer the possibility to provide fibers and textiles with temperature management and thermal balance properties [2]. PCMs have been used for thermotherapy in medical and health care, thermal energy storage, heat pumps, solar engineering, electronic items, thermal triggers, microprocessors, and microclimate control [3,4]. They are functional materials with high latent heat of fusion and crystallization, which melt and solidify at nearly constant temperatures [5]. Thermal energy can be stored in three ways: sensible heat, latent heat, and thermo-chemical heat. Among those, latent heat has been considered the most promising option due to the high density of thermal energy storage and the easy operation at an approximate isothermal condition [6,7]. 

A PCM to be used for latent heat storage applications has to meet certain criteria: (i) Melting and freezing temperatures should be near the intended operation temperature. (ii) High specific heat, high latent heat of fusion, low density, and low vapor pressure at occurring temperatures are beneficial; kinetic properties of PCM, such as rate of nucleation and crystal growth, as well as total crystallinity, are the result of the final latent heat transfer [8]. (iii) Super-cooling should be avoided, since crystallization can be suppressed or completely lost, depending on the PCM’s nature and condition, such as impurity and limited space [9]. (iv) Carrier materials for PCMs should be chemically stable, and the resulting composite should be nontoxic and flame resistant. (v) Moreover, economic considerations, such as high availability at low cost, low environmental impact, no pollution during service life, and potential for recycling, have to be considered with respect to the actual application [10].

The carrier of a latent heat storage composite enables the use of PCMs in a stable state [11,12]. Inorganic porous containers [13], metal foams [14], polymer networks [15], and microcapsules [16,17] are commonly used. For all carriers, high specific surface area, large porous volume, and good thermal stability are basic requirements. Leakage of possibly harmful liquid PCM from the carrier, which could cause trouble in an actual application, has to be prevented [18]. Chemical modifications of carriers have been applied to improve the bonding between PCM and carrier [19]. In general, carrier morphology and dimensions can strongly influence the efficiency of PCMs.

A stable encapsulating structure is a prerequisite to limit the migration of PCMs. In the early 1980s, PCM containing microcapsules were introduced in textiles to improve their thermal performance, intended to be used in astronauts’ space suits in the context of a NASA research program [20]. Meanwhile, a number of studies have been dedicated to spin fibers or to coat fabrics with encapsulated PCM. Coating of fabrics, which can change the properties of a textile in an undesired way, can be avoided if the PCM microcapsules are incorporated within the fibers, e.g., by melt-spinning of a polymer blend, or by bicomponent spinning. Both approaches embed the microcapsules within the fibers, which can thus withstand machine wash. However, this approach requires special microcapsules, and the resulting thermal efficiency is rather low (up to 50 J/g) [21]. 

A high PCM loading is a minimum requirement to achieve high latent heat storage performance. Besides encapsulation technology, filling is another simple approach to produce PCM containing composites. In 1974, Hansen, R. [22] proposed to fill a liquid, in which CO_2_ is dissolved, into hollow fibers to provide thermal insulation. The fibers are meant to improve the insulation ability of a fabric, by gas inflating the tubular fibers when the ambient temperature falls, due to solidification of the liquid that reduces solubility and thus expels the CO_2_. A decade later, Vigo and Frost proposed filling hydrated inorganic salts and PEG into hollow rayon and polypropylene (PP) fibers to achieve temperature-control textiles; they found that the heat capacity of PCM-filled fibers decreased with repeating heating–cooling cycles [23,24]. In this context, the influence of adsorbed water molecules on a shortening of latent heat storage time during melting, on thermal stability, and on thermal behavior should be considered [25]. A vacuum impregnation method to fill paraffin into hollow PP fibers was presented by Zhang et al. [5], where the highest achieved PCM loading ratio was 82 wt.%. With the help of core–sheath electrospinning, PCM nonwovens can be achieved, but limited PCM loading and mass production issues are problems that need consideration [26,27].

Bicomponent (conjugated) spinning technology is one of the most interesting developments in the field of man-made fibers [28]. It can yield flexible fibrous PCMs, which can be used in yarns, nonwovens, woven, and knitted fabrics, and many other application fields [29,30]. Zhang et al. [31] investigated bicomponent fibers with PEG (30–50 wt.%) and thickening agents as core, and PP as sheath; they found that a high viscosity of the core part favored stable fiber melt-spinning, and that fast crystallization improved spinnability by decreasing the leakage tendency of the molten core. Another melt-spun bicomponent PCM fiber type was produced with icosane and a polyolefin blend as core, and PP as sheath [32]. The maximum PCM (icosane) ratio achieved was 21 wt.%, and the resulting fiber achieved a latent heat of 32 J/g, which was only 50–60% of the theoretically possible value, meaning that a significant portion of the PCM in the fiber core did not contribute to the liquid–solid transition.

Hufenus et al. [33,34] developed a novel melt-spinning technology to continuously fill a tubular polymeric fiber with a liquid, where the liquid is injected by a high pressure pump into the polymer melt delivered by an extruder. So far, the highest loading achieved with this approach was a liquid core ratio of about 25 vol.%. To realize a higher core content, first, a thin-walled hollow filament has to be melt-spun through a spinneret with an annular co-flow channel [35] and subsequently filled with the intended liquid by microfluidic technology. Microfluidics provides new ways to produce liquid-core fibers, like post-filled triangular continuous core fibers [36], necklace-like microfibers produced with co-flow technology [37], or kidney-shaped continuous core wet-spun fibers [38]. 

In this paper, PEG was considered as promising thermal energy storage material because of its relatively large heat of fusion, nonproblematic melting behavior, nontoxicity, non-corrosiveness, wide melt-processing window, and long-term stability as PCM [39,40,41]. Polyethylene glycol 1000 (PEG1000) was injected into as-spun hollow PP filaments using a microfluidic setup, and latent heat storage and release, thermal stability, tensile properties, and phase change behavior as a function of fiber drawing were first studied. A similar technique was applied in an earlier study to fill a PP fiber with an ionic liquid containing copper decorated muscovite, thus producing a flexible phase change fiber [42].

## 2. Experimental

### 2.1. Raw Materials

Polypropylene PPH 10099 with a melt flow index of 35g/10min (at 230 °C and a load of 2.16 kg) and a density of 0.905 g/cm^3^, purchased from Total Petrochemicals (Brussels, Belgium), was used to melt-spin a hollow monofilament. PEG1000, a chemical grade PEG with a weight average molecular weight of around 950–1050 g/mol, melting temperature of 33–40 °C, and a density of 1.2 g/cm^3^ (at 20 °C), purchased from Sigma-Aldrich (Buchs, Switzerland), was utilized to fill the hollow filament. Hand knitting yarn, blended from 75% virgin wool and 25% polyamide (PA), purchased from Max Gründl (Ingolstadt, Germany), was used to knit a textile fabric, embedding the PCM-filled filament as carrier matrix for thermal insulation tests.

### 2.2. Sample Preparation

#### 2.2.1. Hollow Filament Preparation

A hollow PP monofilament was produced on a customized pilot-scale melt-spinning plant, originally built by Fourné Polymertechnik (Alfter, Germany), and described in detail elsewhere [43]. Prior to melt-spinning, PP was dried in a vacuum oven at 70 °C for 12 h. Dried PP was melted and pressurized in a single screw extruder with diameter of 14 mm and length to diameter ratio (L/D) of 25. Extrusion temperatures were graduated between 180 and 165 °C, and pressure needed for an effective operation of the melt pump was set to 60 bar. A gear metering pump (Mahr, Göttingen, Germany) was set to a nominal polymer volumetric flow rate of 7.5 cm^3^/min, resulting in a spin pressure of 139 bar and a throughput of 6.79 g/min. To shape the hollow filament, a multiple die with annular co-flow channel [35] was applied, consisting of a tube with 0.6 mm inner diameter and 0.9 mm outer diameter within a 1.2 mm capillary. The polymer melt was spun through the annulus, while pressurized air was injected through the inner capillary tube to assist the formation and stabilization of the hollow core. Finally, the filament was solidified by air quenching at 10 °C and collected as-spun (i.e., without drawing). The resulting hollow filament had a linear density of 0.1 mg/m (1 dtex), and an inner and outer diameter of ~0.85 and ~0.95 mm, respectively [42].

#### 2.2.2. PCM-Filled Filament Preparation

A customized setup was used to fill the hollow PP filament with PEG1000, consisting of a micro-fluidic syringe pump (LSP01-1BH, Longer Precision Pump, Baoding, China) attached to the hollow filament, using a soft silicon tube as connection, and epoxy resin as sealing adhesive.

First, PEG1000 was molten in a sealed container at 60 °C. The thus liquefied material was then injected into the hollow filament, performing the filling in an environment temperature of 50 °C, in order to achieve PCM-filled PP filament samples. After filling, both ends of the core–sheath filaments were heat-sealed to avoid leakage of PEG1000.

The filled filaments were subsequently drawn on a Zwick Z100 (ZwickRoell, Ulm, Germany) tensile testing machine to reduce diameter and improve tensile properties (cold drawing). To this end, a 5 cm long PEG1000-filled PP fiber sample was fixed between two pneumatic clamps. Drawing was performed in a heating chamber at 40 °C with a crosshead speed of 20 mm/min, and with draw ratios (DR) of 2.0, 2.5, and 3.0, respectively. 

#### 2.2.3. PCM-Enhanced Knitted Fabric Preparation

Textile fabric samples (2.5 times 2.5 cm) were prepared for infrared thermal image analysis by hand-knitting, using wool–PA blended yarn as base material, and PCM-filled PP fibers as inlay inserted in every second knit row (Figure 1).

### 2.3. Characterization

#### 2.3.1. Morphology 

The macro-scale morphology of PEG1000-filled PP fibers was observed with the digital microscope VHX-1000 (Keyence, Neu-Isenburg, Germany). All samples were tested without further treatment.

The micro-scale morphology of the inner surface of hollow PP fibers before and after contact with PEG1000 were observed with the scanning electron microscopy (SEM) Hitachi S-4800 (Hitachi, Tokyo, Japan). The acceleration voltage was 2000 V at a current of 10 µA. Before SEM analysis, the samples were Au-coated in vacuum. 

#### 2.3.2. Thermal Properties 

Thermal properties were characterized using differential scanning calorimetry (DSC) and thermogravimetric analysis (TGA). Enthalpy of melting and crystallization were analyzed with the DSC 214 Polyma (Netzsch, Selb, Germany) under N2 atmosphere at a flow rate of 40 mL/min. The DSC was calibrated using a blank aluminum pan. First, the sample was heated from room temperature to 60 °C, followed by an isothermal step of 5 min to eliminate thermal history, then a cooling ramp down to −40 °C and another isothermal step of 5 min, and finally a second heating ramp to 60 °C again. Heating and cooling rates were set to 10 °C/min. To avoid an effect of air trapped within PEG1000-filled fibers on the melting behavior of the PCM, only samples free from bubbles inside the fiber were selected for the tests. Thirty cycles of above DSC procedure were applied to test the thermal reversibility of PCM fibers. 

To characterize the thermal stability of PCM fibers, TGA (TG 209 F1, Netzsch, Selb, Germany) was performed under nitrogen atmosphere (flow rate 20 mL/min), increasing the temperature from 25 to 650 °C at a heating rate of 10 °C/min.

#### 2.3.3. Thermal Infrared Image Analysis

The infrared camera IngraTec Vh-780 (InfraTec, Dresden, Germany) was utilized to catch thermal patterns and to derive heat release curves of fabric samples. First, a test sample was exposed to a temperature of 40 °C (infrared lamp) for 20 min, and then it was quickly transferred to an isothermal environment of 23 °C (by switching off the lamp) in order to collect infrared images. 

To prevent curling of the knitted samples, and to provide good contact with the test platform, the fabric was fixed on a custom-designed sample holder. All images were analyzed by converting pixel information to heat release curves with the image processing software that comes with the test instrument. 

#### 2.3.4. Mechanical Properties

Tensile properties of filaments were measured with the automatic tensile tester Statimat ME+ (Textechno Herbert Stein, Mönchengladbach, Germany), using a 100 N load cell. First, a preload of 1 cN/tex was applied to straighten the fiber. The starting free fiber length was 50 mm, and the crosshead speed was set to 200 mm/min. For each fiber type, tests were repeated 10 times, and mean value and standard deviation were calculated. Before the test, all samples were conditioned at a room temperature of 23 °C and a relative humidity of 50%.

#### 2.3.5. Chemical Characterization 

Attenuated total reflection Fourier transform infrared (ATR-FTIR) spectra were recorded with the spectrometer Vector 33-MIR (Bruker Optik, Ettlingen, Germany) in the wave number range of 500 and 4000 cm^−1^.

## 3. Results and Discussion

### 3.1. PCM Loading Ratio of PEG1000-filled PP Fibers

The phase transition enthalpy of PEG1000-filled PP fibers strongly depends on the core-sheath ratio and thermal storage capacity of the PCM core of the bicomponent fiber. 

The volume fraction *V_core_* of PEG, with respect to a fully filled fiber, can be calculated based on the inner and outer diameter of the hollow PP fiber, as shown in Equation (1). Considering the densities of PEG1000 and PP, the weight percentage *W_core_* of PEG1000, regarding the filled PP fiber, can be deduced according to Equation (2).
(1)$$V_{core}=\frac{D_{inside}^{2}}{D_{inside}^{2}+D_{outside}^{2}}\times 100\%$$
(2)$$W_{core\;}=\frac{\rho_{core}D_{inside}^{2}}{\rho_{core}D_{inside}^{2}+\rho_{sheath}\left(D_{outside}^{2}-D_{inside}^{2}\right)}\times 100\%$$
where *D_inside_* is the core diameter and *D_outside_* is the overall diameter of the bicomponent fiber. *ρ_core_* and *ρ_sheath_* are the densities of PEG1000 (1.128 g/cm^3^) and PP (0.905 g/cm^3^), respectively. The hollow PP filament has average inner and outer diameters of *D_inside_* ~ 0.85 mm and *D_outside_* ~ 0.95 mm, respectively (Figure 2). As a result, the PCM volume fraction of the fiber can be calculated as ~80 vol.%, and the respective weight fraction as ~83 wt.%. These values exceed those achieved by Deng et al. [6], where composites of PEG, expanded vermiculite and silver nanowires were utilized for thermal energy storage, and the highest PEG loading achieved was ~66 wt.%. In the study by Zhang et al. [5], where vacuum impregnation was applied to fill hollow PP fibers with paraffin, the highest PCM loading achieved was ~82 wt.%. However, other than in our present study, residual PCM outside of the fibers could not be avoided. 

### 3.2. DSC Analysis

Latent thermal energy storage and release by PCMs can be related to the enthalpy change during melting and cooling, measured by DSC. Figure 3 and Table 1 show DSC results of pure PEG1000, as-spun hollow PP filament, and PEG1000-filled PP fibers (both as-spun and with DR = 2).

In Figure 3, no peak can be found in the PP curve within the relevant phase change temperature range of PEG1000 (~0–50 °C), while PEG1000 and the PEG1000-filled PP fibers showed evident melting and crystallization peaks (Table 1). The peak melting temperatures of PEG1000-filled PP fibers (80 vol.% PEG content) were a little higher than those of pure PEG1000 and remained about the same for post-drawn fibers. The peak crystallization temperatures of PCM-filled fibers, on the other hand, were a little lower than those of pure PEG1000, but they increased when the fibers were post-drawn. 

Encapsulation ratio (*R*) and encapsulation efficiency (*E*) can be deduced from the values of enthalpy of PEG1000 and PEG1000-filled PP fibers according to Equations (3) and (4) [44]:(3)$$R=\frac{\Delta H_{m,\;PEG1000\;filled\;PP}}{\Delta H_{m,\;PEG1000\;}}\times 100\%$$
(4)$$E=\frac{\Delta H_{m,\;PEG1000\;filled\;PP@\mathrm{Normalized}\;\;}+\Delta H_{C,\;PEG1000\;filled\;PP\;@\mathrm{Normalized}\;}}{\Delta H_{m,\;PEG1000\;}+\Delta H_{c,\;PEG1000\;}}\times 100\%$$
where *ΔH_m_* and *ΔH_c_* are enthalpies of melting and crystallization, respectively.

Table 1 summarizes all measured and calculated thermal properties. It can be seen that the normalized enthalpy of PEG1000-filled PP fiber, which only considers the PCM encapsulated within the fiber, is lower than that of pure PEG1000, and that the cooling enthalpies are below those of melting. Considering different PEG contents in undrawn PCM-filled PP fibers, the normalized enthalpies for melting and crystallization showed similar tendencies for all tested samples. Post-drawing, on the other hand, reduced the enthalpies, as shown in Table 1.

For PEG1000-filled PP fibers, the filled core ratios are 80.3 and 80.2 wt.% for undrawn and drawn fibers, respectively, with respective filling efficiencies of 96.7 and 93.7 wt.%. This means that the drawn fibers show a lower encapsulation efficiency. Compared to other studies, Zhang et al. [5] prepared flexible phase change composite fibers with loading ratios as high as 82%, and Luo et al. [45] prepared flexible paraffin/Multi-wall carbon nanotubes/PP hollow fiber membranes with a maximum paraffin absorption capacity of 52%.

The polymer crystallization process involves primary nucleation and secondary crystal growth [46]. For PEG, the crystallization behavior is a prerequisite for storage and release of PEG’s latent heat, which is affected by the interaction between PEG molecules and the internal surface of the containment [47]. PEG crystal growth is a thermo-dynamically feasible process [48]. The primary nucleation is triggered by thermal fluctuations of polymer chains at an early stage of polymer crystallization [49], followed by continuous growth of the primary nucleus to form a stable crystalline phase [50,51]. The lack of fluctuation of PEG chains will slow down its crystallization, but this tendency is related to both, the main chain itself, and its surroundings. The former is a physical influence, while the latter is related to an interaction between containment and PCM. 

The phase change entropy of PEG1000 inside tubular PP fibers is lower than that of pure PEG1000, meaning that in the former case, PEG1000 does not crystallize completely, which can be explained by uneven crystals or smaller spherulites. This would also explain why the onset temperature of melting of PEG1000 drops when encapsulated for most examples. However, filling ratios and filling efficiencies are between 33% and 42% higher compared to elsewhere reported values for a PCM containing 50 wt.% PEG1000 within diatomite [52]. 

In another study, PCM was confined within a porous carrier, where no interconnected uniform crystalline domain could form, showing that the crystallization process took place in local domains separately [53]. In our case of PEG1000-filled fibers, the PCM formed a continuous phase, giving the material the chance to crystalize as a whole, and thus to form a perfect crystal, which results in higher enthalpy of melting and crystallization. However, the phase change process is also influenced by intermolecular incidents like physical entanglement, as has been observed in fibers melt-electro spun from mixtures of polyurethane (PU) PCM and polymethyl methacrylate (PMMA) [54]. Here, the actual phase-change enthalpies of the mixtures were considerably lower than the corresponding theoretical values, most likely because the PU-PCM molecules did not have enough time to form well-defined crystallites upon cooling, since the surrounding PMMA molecules limited their movement during the crystallization process [54]. 

The motion pattern of molecular segments within a PEG1000 matrix differs from the one at the interface between the PCM and the PP sheath. The latter will be influenced by both the interfacial tension and the specific interface area between PEG1000 and PP. The thinner the fiber is, the higher this specific area will be. Considering those two factors, we prepared partially filled PP fibers with different core ratios, to study the effect of the specific interface area on the phase change behavior of the PCM-filled fibers, see Figure 4 and Table 1.

Elsewhere reported results for storage material with narrow porous structure indicate that the PCM within a composite might not endure the expected thermal storage-release cycles, since restricted mobility of the organic PCM macromolecules, confined by capillary forces, decreases their endothermic and exothermic enthalpy [5,55]. In the case of PCM-filled filaments, the encapsulation efficiency can be increased by increasing the PEG content, which could easily be achieved by a continuous filling process [33]. However, as can be deduced from Table 1, the encapsulation efficiency drops with increasing draw ratio of PCM-filled fibers, which can be explained by the combined effects of higher specific PEG–PP interfacial area and higher percentage of free molecular chains, which lead to an uneven and imperfect crystallization of PEG1000.

Wang et al. [13] reported that grafting amino groups on the internal surface of mesoporous silica SBA-15, utilized as PCM support, can reduce the hydrogen bond interactions between PEG molecules and the channel surface of SBA-15, thus altering the adsorption conformation of PEG chains from train to loop structure, which favors their stretching and crystallization. However, similar polarities of PEG molecules and silanol groups led to spillover of PEG onto the external surface of the support, thus suppressing crystallization of the PCM.

The hydrophobic PP has a surface tension of 30.9 mN/m, while the surface tension of the hydrophilic PEG is 37.9 mN/m [56]. As a result, PEG will not adsorb on the inner surface of the PP fiber sheath, and the crystallization of PEG1000 will not be influenced by molecular chains that adhere to the inner surface of the tubular fiber. This explains why the enthalpy of both fusion and solidification of PEG1000 showed no evident change compared to PCM-filled PP fibers. The reason the crystallization enthalpy decreased for drawn fibers is their increased specific PEG–PP interface, which can promote spreading of PEG on the inner surface of the PP fiber. The tendency of imperfect crystal formation especially increased near bubbles that formed during incomplete PCM filling, as shown in Figure 5. The apparently transparent areas within solidified PEG1000 are most probably non-crystalline or low-crystallinity domains.

The heat storage efficiency of PCM is expressed by the heat loss percentage, *E_Lost_*, defined as follows:(5)$$E_{Lost\;}=\left(1-\frac{\Delta H_{c}}{\Delta H_{m}}\right)\;\times \;100\%$$

Here, *ΔH_c_* is the solidification or crystallization enthalpy, and *ΔH_m_* is the fusion or melt enthalpy, stemming from DSC [39].

*E_Lost_* for PEG2000, combined with NH_2_-SBA-15-CH_3_, was found to be 6.8% [13], combined with an epoxy resin ~7.2% [57], and combined with a polyurethane membrane 11.1% [58]. In comparison, *E_Lost_* of the PEG1000-filled PP fibers was 7.7% in the as-spun state, and 1.5% in the drawn state (DR = 2).

During the drawing process, bubbles at the PEG1000–PP interface are expected to stretch due to slippage of PEG molecular chains on the incompatible PP surface, as shown in Figure 6. PEG will not adsorb on the inner surface of the tubular PP filament and thus not spread as a layer (loop structure), which restrains crystallization of PEG1000. Higher draw ratios result in a small decrease of enthalpy. The microscope pictures reveal a distinct boundary between PEG and the inner surface of the PP fiber sheath. Under the drawing force, a bonding point of PEG on PP may slip along the surface, resulting in the development of an internal pore (bubble). Those bubbles in the core part move along the core–sheath interface and break up into smaller ones, while new pores develop. In consequence, the gas PEG1000 interface area increases, which leads to the small decrease in the crystallization of PEG1000 upon cooling, as seen in Table 1.

### 3.3. Thermal and Chemical Stability

The thermal stabilities of pure PEG1000, hollow PP fibers and PCM-filled PP fibers are shown in Figure 7.

The 50% weight loss temperature of pure PEG1000 is 380.4 °C, and the final weight loss at 385.5 °C is nearly 96.6 wt.%. For PEG1000-filled PP fibers, two weight loss steps can be observed, a first step of 84.7% (with a 50% weight loss temperature at 384.7 °C), and a second of 10.4% (onset at 416.8 °C), corresponding to the degradation of PEG and PP, respectively. In comparison to pure PEG1000, the weight loss of PEG1000 in the PP fiber is proceeding slower, meaning that the PP container slightly enhances the PCM’s thermal stability [5].

To study the chemical stability of both PEG1000 and PP when combined in a bicomponent filament, fiber samples, stored for two years, where subjected to ATR-FTIR analysis (Figure 8).

From Figure 8, we can find that 2950 cm^−1^ and 2868 cm^−1^ are the asymmetric and symmetric stretching vibrations of CH_3_, and 2916 cm^−1^and 2838 cm^−1^ represent the asymmetric and symmetric stretching vibrations of CH_2_, respectively [59]. Furthermore, 1106 cm^−1^ represents the stretching vibration of the ether bond [60]. No relevant peak shifts of PP could be observed, meaning that no chemical change of the PP occurred under prolonged contact with PEG1000, and that the bicomponent system is chemically stable.

### 3.4. Thermal Infrared Imaging

The PEG1000-filled PP fibers with up to 83.3 wt.% of PCM core naturally exhibited a thin sheath, which rendered the filament mechanically weak. This impeded standard knitting or weaving into fabrics to produce a test sample, or to enable garment applications. In consequence, a supporting yarn was utilized to prepare knitted fabrics containing a certain amount of PCM-filled fibers.

The actual PEG1000 content, *W_PCM_*, of suchlike prepared test samples can be calculated as
(6)$$W_{PCM}\;\%=\;\frac{W_{2}\times W_{core}}{W_{1}}\times 100$$

Here, *W*_1_ is the total weight of the knitted fabric, *W*_2_ is the weight of the PCM-filled PP fibers, i.e., *W*_1_ minus the overall weight of the supporting wool/PA yarn within the fabric, and *W_core_* is the weight percentage of PEG1000 within the filled fibers.

Thermal energy storage (TES) is the temporary storage of excess high or low temperature energy for later use, which bridges the time gap between energy requirements and usage [61]. The consequential insulation effect of PCM is dependent on temperature and time. When the environment temperature rises to the phase change temperature, i.e., the material’s onset melt temperature, the PCM absorbs heat while liquefying, and stores this energy as latent thermal resource. When the temperature falls, the PCM solidifies and releases the stored heat upon phase conversions. This process starts at the onset solidification point of the PCM and it terminates when all PCM has solidified or crystallized, meaning that this process can be considered as a kind of dynamic thermal insulation. The respective thermal insulation capability is related to the amount of PCM used, since it depends on the enthalpy of melting and cooling of the material. 

When a PCM reaches its phase change temperature (melting or crystallization point), it absorbs or releases heat almost isothermally until the transition is complete, which motivates its applicability in thermal insulation [62]. In case of the knitted wool–PA fabric samples containing 10.2 wt.% PEG1000-filled PP fibers, the overall PCM content is only 8.5 wt.%. Nevertheless, an evident thermal release peak can be found in the temperature gradation curves gained from infrared imaging (Figure 9). Compared to the reference fabric without PEG1000-filled PP fibers, the PCM-enhanced sample showed a temperature preservation plateau starting at 26.1 °C, and persisting at around 25–26 °C for ~5 min. It took the PCM-enhanced fabric ~18 min longer than the reference sample to reach the final temperature of 20 °C. The observed temperature progression is in agreement with the DSC results.

Figure 9 also reveals that the starting temperature of the heat release curve of the sample containing PEG1000-filled PP fibers is higher than that of the reference fabric, meaning that the thermal conductivity of the former is lower than that of the latter. 

In general, a latent heat storage system should contain at least the following components: a material that undergoes a solid to liquid phase transition within the desired operating temperature range, and that can store added heat as the latent heat of fusion (in our case PEG1000); a containment for the energy storage substance (in our case the tubular PP fiber); a surface that enables a heat transfer from the source to the carrier. The respective components can be assigned to two groups: heat storage materials and heat exchangers [63]. Thus, it is of interest how crystallization happens in the PCM within the tubular PP fiber during the phase change process (plateau in Figure 9). To answer this question, an optical microscopy investigation was performed (Figure 10).

From Figure 10, it can be seen that crystallization began at both ends of the fiber, and that the crystal growth propagated along the fiber until the core of the fiber was fully crystallized (heat release and crystallization are intertwined). At the same time, it can be observed that bubbles form within the molten PEG1000 and that the boundaries of the bubbles become irregular over time. This transformation does not happen while crystals grow, but only after the PCM core has completely solidified, because the air gap strongly reduces thermal conductivity.

To observe the PCM melting behavior, we used optical microscopy to trace crystalline changes of PEG1000 within the tubular PP filament after short (up to 20 s) local heating at 45 °C (Figure 11). The respective pictures reveal transparent spots near the boundaries of the molten parts; some tiny crystals can also be found. Thus it can be concluded that, during short local heating, PEG1000 only melts at the core–sheath interface, where it is in direct contact with PP. When further maintaining the elevated temperature, the molten PEG domains spread along the interface until the test temperature reached the solidification temperature of PEG1000.

### 3.5. Thermal Reversibility

The thermal reversibility is a key factor in assessing PCMs. Thermal performance under cycling testing conditions, evaluated with DSC, is depicted in Figure 12. It can directly be concluded that PEG1000-filled PP fibers have a good thermal reversibility. Figure 13 shows the evolution of enthalpies, as well as onset and peak temperatures of the PCM-filled fibers as a function of endothermic and exothermic cycling (30 times) in DSC. 

It can be seen that the latent heat of fusion of PEG1000-filled PP fibers slightly varied but stayed in a range of about ±1%, while the latent heat of crystallization revealed a gradual drop. The peak melting temperature was stable with a progressive number of thermal cycles, but the crystallization peak temperature increased gradually. The onset temperature decreased by trend during both endothermic and exothermic cycling. This increasing solidification temperature can best be explained by a slight improvement of the crystallization of PEG1000, a tendency which was also reported for PEG1000-diatomite PCM, where the onset melting temperature increased by 1.94 °C, the freezing temperature by 0.41 °C, the latent heat of fusion by 1.1%, and the latent heat of crystallization by 3.5% after 1000 thermal cycles [52]. 

Figure 14 reveals no evident change in onset temperatures of melting and cooling of PEG1000 after 30 cycles, meaning that the phase change temperature does barely vary during the cycles, and that the PCM stays thermally stable and reversible within the PP fiber. The same tendency was also reported by Sharma et al. for PEG6000 after 1500 cycles [41].

### 3.6. Tensile Properties

Good tensile properties are a key requirement for textile fiber applications. To mimic the online production of liquid-core melt-spun PCM fibers, as-spun hollow PP filaments were filled with PEG1000, and subsequently cold-drawn in a tensile testing machine. This enabled studying both fiber drawability and tensile properties. 

For tensile testing, PEG1000-filled PP fibers with DRs of 2.0, 2.5, and 3.0 were prepared. Due to the rather thin polymeric sheath of the bicomponent fiber, applying a higher DR than 3.0 resulted in breakage of the filled fibers. As shown in Table 2, both ultimate tensile stress and Young’s modulus increase with draw ratio, while ultimate tensile strain decreases, which is caused by progressive orientation of the PP macromolecules upon drawing. However, the toughness, i.e., the work of fracture, strongly decreases for drawn fibers, which is most probably due to splitting and fibrillation of the thin polymeric sheath under strain (Figure 15), which is atypical for standard PP filaments. The mechanism of fiber breakage was further investigated by SEM, revealing the presence of spherulitic crystalline structures in the PP sheath (Figure 16), which are known to promote brittleness and fibrillation. Due to their dimensions of several micrometers, their impact is augmented by a thinning of the filled fiber’s polymeric sheath, which goes along with increasing the draw ratio.

## 4. Conclusions

PEG1000, commonly used for latent heat storage applications, was utilized to produce PCM-filled PP fibers, prepared by hollow filament melt-spinning and subsequent microfluidic injection. The obtained bicomponent PEG-PP fiber reached a maximum core content of 83 wt.%, and the respective thermal efficiency of the undrawn PCM fiber was 97%, proving that the PP sheath had basically no effect on the phase change behavior of the confined PEG1000. The formation of bubbles in the fiber core could not be prevented; in consequence, imperfect PEG crystals formed near those bubbles, thus affecting the crystallization behavior of PEG1000. It was found that the fewer the bubbles, the higher the thermal efficiency. Crystallization of PEG1000 upon cooling predominantly started at the fiber tips and at the inner surface of the PP sheath, most probably because the PP interface acts as nucleating site for the molten PEG. Upon further solidification of the core, crystallization proceeds to the inside of the fiber until crystallization of PEG1000 is complete. In the case of a local heat source, the PCM core only melts in the vicinity of the source, while the rest of the fiber core remains crystalline. To achieve a complete phase change from solid to liquid, an extensive heat source is required.

Stretching PEG1000-filled PP fibers by cold-drawing was found to lower the thermal efficiency as a function of increasing DR. This can be explained by a reduced diameter of the bicomponent fiber, implicating an increased specific core–sheath interface area, i.e., PEG-PP contact area. The phase change enthalpy of the PCM-filled fibers remained basically unchanged after 30 heating–cooling cycles. A thermal insulation performance test revealed a significant heat release peak of knitted wool–PA fabrics containing 10 wt.% PEG1000-filled PP fibers, as well as a distinct temperature plateau starting at the onset of PEG crystallization and remaining at 25–26 °C for about 5 min. The cold-drawn PCM-filled PP fibers also showed sufficient tensile properties for careful textile processing, in particular maximum Young’s modulus and ultimate tensile stress of 2.6 and 0.25 GPa, respectively.

## Figures and Tables

**Figure 1 materials-14-00401-f001:**
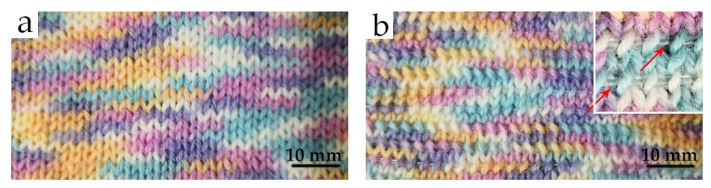
Knitted wool–polyamide (PA) fabrics (**a**) without and (**b**) with inserted phase change material (PCM)-filled polypropylene (PP) fibers.

**Figure 2 materials-14-00401-f002:**
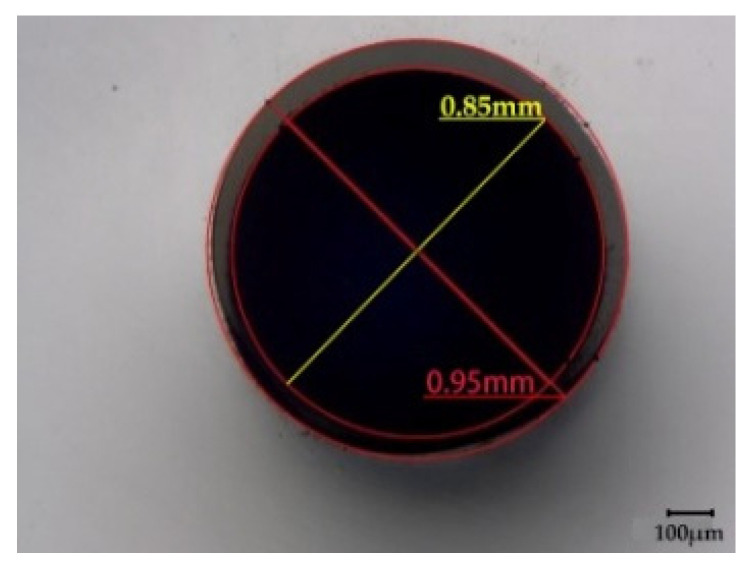
Cross-section of the as-spun hollow PP filament.

**Figure 3 materials-14-00401-f003:**
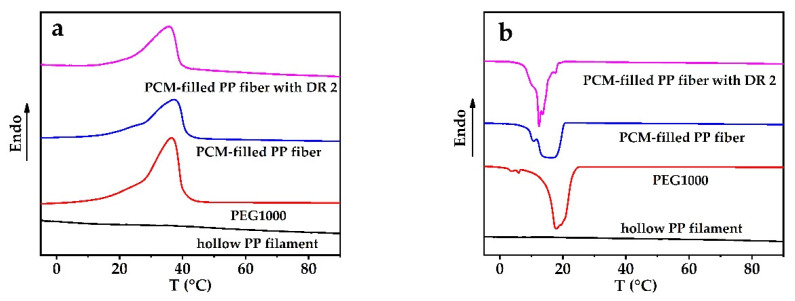
DSC curves of polyethylene glycol 1000 (PEG1000), hollow PP filament, PEG1000-filled as-spun PP fiber(PCM-filled PP fiber), and PEG1000-filled PP fiber, cold-drawn at room temperature to a draw ratio (DR) of 2(PCM-filled PP fiber with DR 2). (**a**) Melting and (**b**) crystallization curves of first heating and cooling cycles, respectively.

**Figure 4 materials-14-00401-f004:**
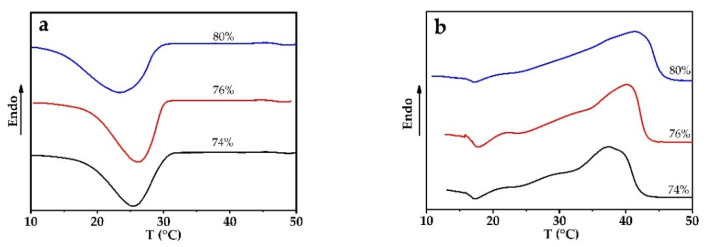
DSC curves of PEG1000-filled PP fibers with filling ratios of 74, 76, and 80 vol.%. (**a**) Melting and (**b**) crystallization curves of first heating and cooling cycles, respectively.

**Figure 5 materials-14-00401-f005:**
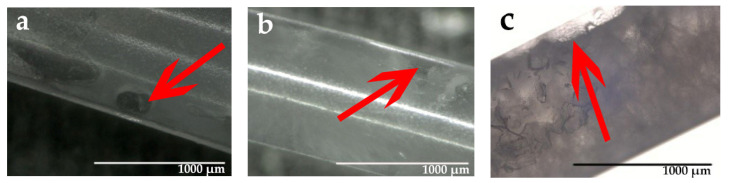
Microscopic pictures of bubbles (indicated by arrows) that formed within PEG1000 as a consequence of incomplete filling of hollow PP fibers. Micrographs (**a**,**b**) show examples of undrawn PCM-filled fibers, (**c**) depicts filled fibers drawn with a DR of 2.

**Figure 6 materials-14-00401-f006:**
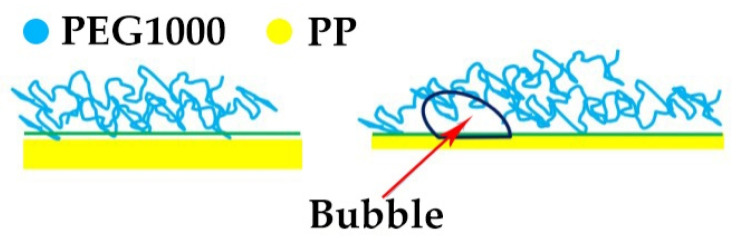
Proposed mechanism of bubble formation at the interface of PEG and the inner wall of the tubular PP fiber under drawing.

**Figure 7 materials-14-00401-f007:**
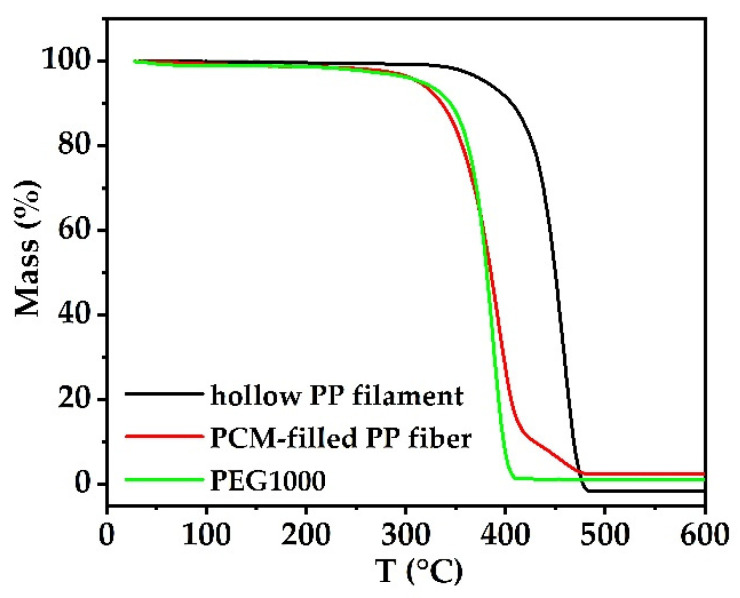
Thermogravimetric analysis (TGA) of hollow PP filament, PCM-filled PP fibers, and PEG1000.

**Figure 8 materials-14-00401-f008:**
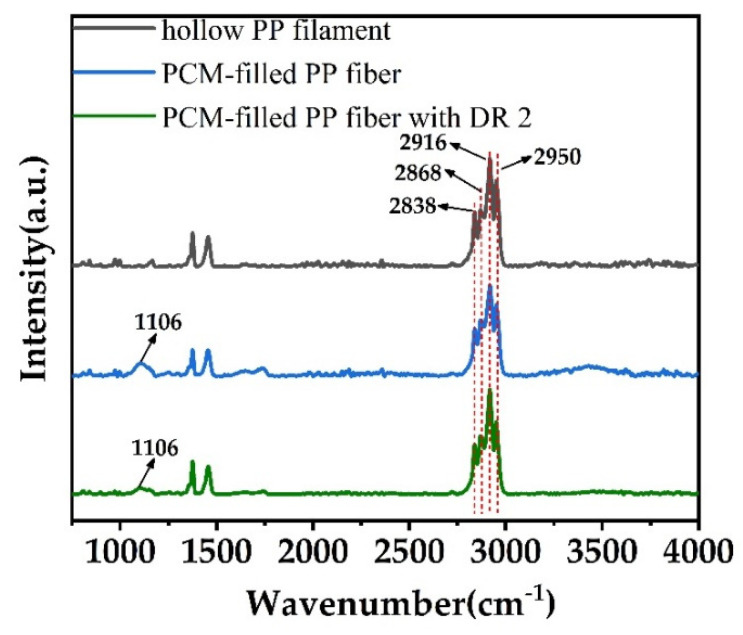
ATR-FTIR curves of as-spun hollow PP filament, and PEG1000-filled PP fiber without and with drawing (DR = 2).

**Figure 9 materials-14-00401-f009:**
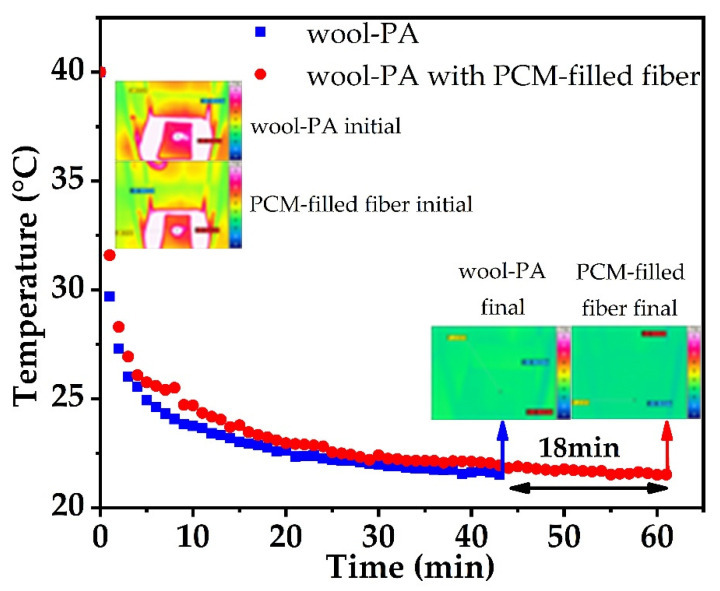
Temperature gradation curves of knitted wool–PA fabric with and without PEG1000-filled PP fibers.

**Figure 10 materials-14-00401-f010:**
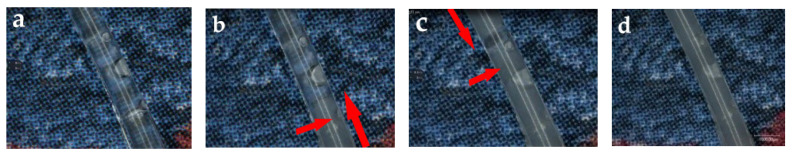
Optical microscopy images of PEG1000-filled PP fibers during cooling, taken after (**a**) 0 min, (**b**) 3 min, (**c**) 4 min, and (**d**) 5 min. Red arrows indicate frozen areas and their spreading tendency.

**Figure 11 materials-14-00401-f011:**
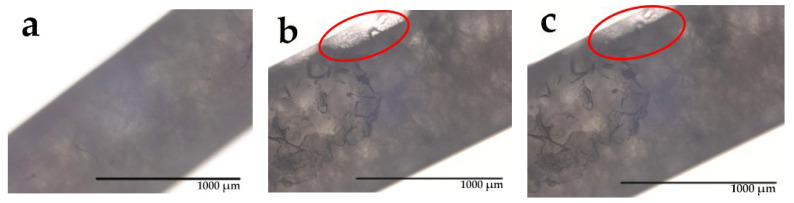
Optical microscopy images of PEG1000-filled PP fibers during local heating, taken after (**a**) 0 s (i.e., before heating), (**b**) 10 s, and (**c**) 20 s. Red ovals indicate spreading molten areas.

**Figure 12 materials-14-00401-f012:**
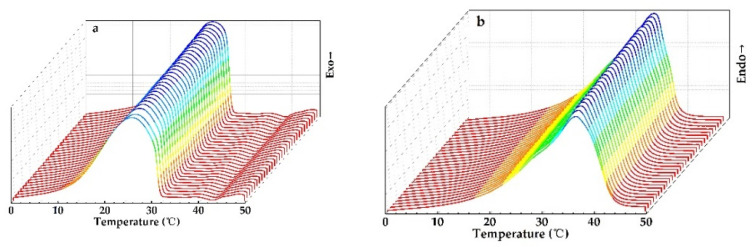
DSC curves of PEG1000-filled PP fibers for 30 (**a**) cooling and (**b**) reheating cycles.

**Figure 13 materials-14-00401-f013:**
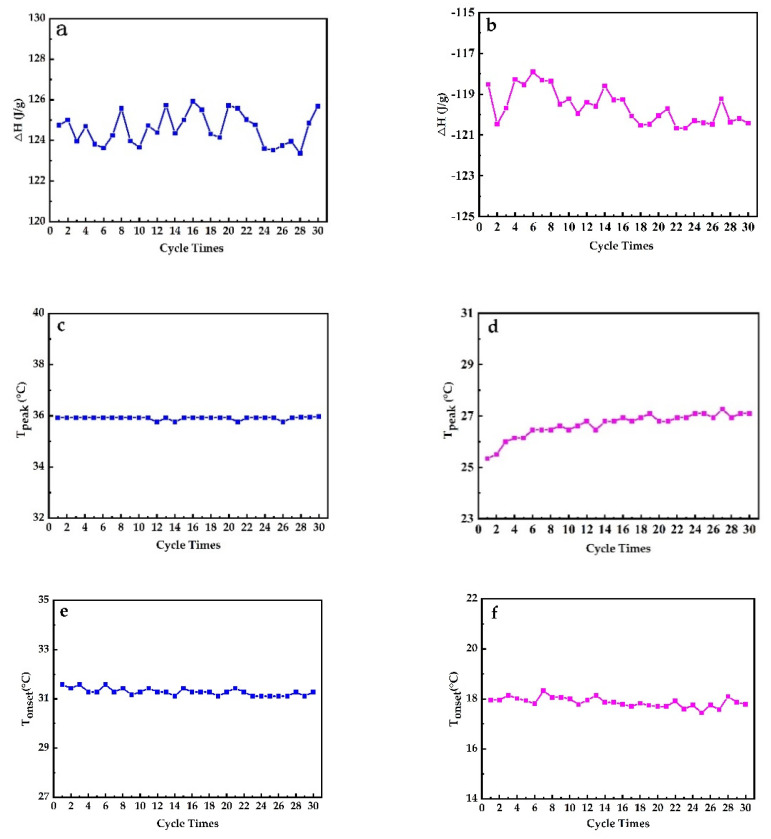
Evolution of (**a**) enthalpy of melting, (**b**) enthalpy of crystallization, (**c**) peak melting temperature, (**d**) peak crystallization temperature, (**e**) onset melting temperature, and (**f**) onset crystallization temperature during 30 heating–cooling DSC cycles performed on PEG1000-filled PP fibers.

**Figure 14 materials-14-00401-f014:**
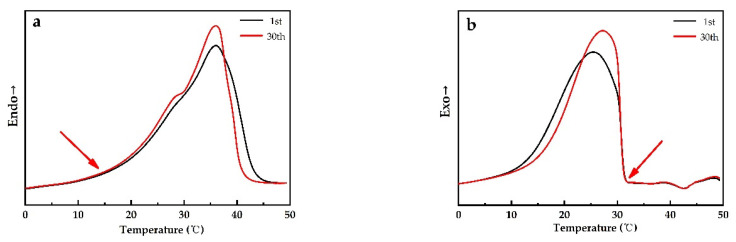
Melting and solidification curves of PEG1000-filled PP fibers at the first and the thirtieth (**a**) melting and (**b**) crystallization curves of first heating and cooling cycles, respectively.

**Figure 15 materials-14-00401-f015:**
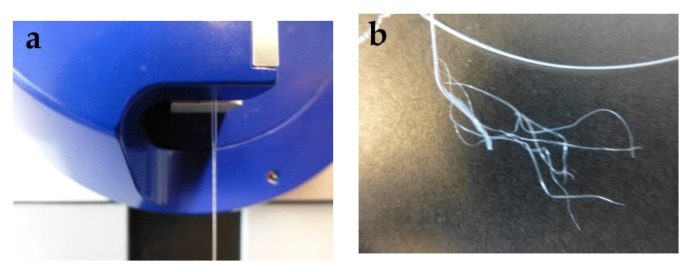
(**a**) Cold-drawing of PEG1000-filled PP fibers, and (**b**) subsequent splitting and fibrillation of the fiber sheath.

**Figure 16 materials-14-00401-f016:**
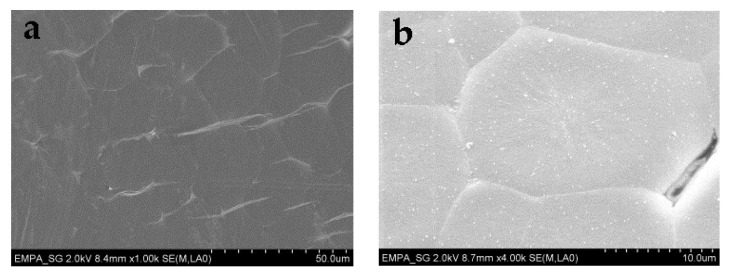
SEM pictures of the hollow PP fiber sheath in two different zoom factors: (**a**) 100×; (**b**) 400×.

**Table 1 materials-14-00401-t001:** Thermal properties of PEG1000 and PEG1000-filled PP fibers, evaluated with DSC. ΔH, R, and E are transition enthalpy, encapsulation ratio, and encapsulation efficiency, respectively. The normalized enthalpy considers the actual weight percentage of PEG1000 within filled PP fibers. Draw ratio DR = 1.0 corresponds to the as-spun, undrawn filament.

Sample	PEG Content (vol.%)	DR	Step	T_peak_ (°C)	T_onset_ (°C)	T_end_ (°C)	ΔH (J/g)		R (%)	E (%)
							Tested	Normalized		
PEG1000	100	-	cooling	18.0	22.4	15.1	132.1	-	-	-
			heating	36.4	25.2	39.7	134.9	-		
PEG1000/PP fiber	80	1.0	cooling	15.9	20.4	11.1	106.7	128.1	80.3	96.7
			heating	37.3	23.0	41.2	108.4	130.1		
	76	1.0	cooling	12.5	15.5	10.2	100.1	120.2	72.7	94.4
			heating	35.5	21.7	39.6	108.3	130.0		
	74	1.0	cooling	26.2	28.9	18.8	88.4	119.5	72.4	91.7
			heating	40.2	29.9	43.1	98.2	132.6		
	80	2.0	cooling	25.4	28.7	18.0	88.3	116.2	80.2	93.7
			heating	37.9	30.3	42.1	97.8	128.6		
	80	2.5	cooling	23.0	30.2	5.5	96.1	115.4	76.4	89.5
			heating	37.2	15.5	45.0	103.0	123.7		
	80	3.0	cooling	22.1	29.9	4.1	100.2	120.3	75.7	90.9
			heating	37.4	17.4	47.0	102.1	122.6		

**Table 2 materials-14-00401-t002:** Tensile properties of as-spun and drawn PEG1000-filled PP fibers. DR = 1.0 corresponds to the as-spun, undrawn filament. MV = mean value, SD = standard deviation.

DR	Ultimate Tensile Stress (MPa)		Ultimate Tensile Strain (%)		Young’s Modulus (Gpa)		Toughness (Work of Fracture) (cNm)	
	MV	SD	MV	SD	MV	SD	MV	SD
1.0	103	19	219	93	0.78	0.07	84	42
2.0	211	14	41	15	2.27	0.09	18	11
2.5	230	16	40	13	2.59	0.16	17	8
3.0	248	28	37	16	2.64	0.15	15	8

## Data Availability

Data is contained within the article.

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
