# Peer review of "Flexible Phase Change Material Fiber: A Simple Route to Thermal Energy Control Textiles"

_materials, 2021, doi:10.3390/ma14020401_

Round 1
Reviewer 1 Report
- This paper provides an excellent investigation on developing thermal energy control textiles
- English needs some improvement.
- Please see some minor changes suggested in the attached manuscript
- The key change needed in the manuscript is that at the end of each result subsection, write briefly the practical implication of the results found.

Author Response
- This paper provides an excellent investigation on developing thermal energy control textiles
A: We thank the reviewer for the positive feedback and we addressed the concerns as follows.
- English needs some improvement.
A: We thoroughly read the text again and made some minor corrections.
- Please see some minor changes suggested in the attached manuscript
A: We changed the marked words and sentences according to the suggestions:
- "polyethylene glycol (PEG) 1000" has been changed to "polyethylene glycol 1000 (PEG 1000)"
- "DSC" has been changed to "Differential scanning calorimetry (DSC)"
- "proofed" has been changed to "proved"
- "Cycling heating-cooling tests proved the reversibility of latent heat release and storage properties of the PCM fiber" has been changed to "Cycling heating-cooling tests proved the reversibility of latent heat release and storage properties, and the reliability of the PCM fiber".
- "phase change behavior as a function of fiber drawing were studied" has been changed to "phase change behavior as a function of fiber drawing were first studied"
- "recorded with the spectrometer Vector 33-MIR (Bruker Optik, Ettlingen, Germany)" has been changed to "recorded with the spectrometer Vector 33-MIR (Bruker Optik, Ettlingen, Germany) with a scan range between 4000 and 500 cm-1.
A: We modified the indicated Figures 1b, 2 and 6 according to the suggestions:
The color of the numbers is deepened. Considering the direct details inside the photos is more easy to read, we did not change label in Figure 2.)
- You need to explain the as-spun and post drawn fibres beforehand.
A: These two words are very common in the field of fiber melt spining, thus we believe that no further explanation is needed here.
- The key change needed in the manuscript is that at the end of each result subsection, write briefly the practical implication of the results found.
A: We added respective explanations to the manuscript, and we marked the changes in red.

Reviewer 2 Report
The manuscript “Flexible Phase Change Material Fiber: A Simple Route to Thermal Energy Control Textiles” addresses an interesting topic and describes clearly the experimental methods and results.
Therefore, it may be recommended for publication after minor revision:
“To study the chemical stability of both PEG1000 and PP when combined in a bicom-368 ponent filament, fiber samples, stored for two years, where subjected to ATR-FTIR analy-369 sis (Figure 8)”
- Authors should improve Figure 8: add important peaks and describe functional groups;
- Authors should broaden the discussion of the chemical characterization results (lines 368-276);
- The following publication is recommended in order to fulfill this section: - Materials Science and Engineering: C, 2019, 100: 837-844.
Author Response
The manuscript “Flexible Phase Change Material Fiber: A Simple Route to Thermal Energy Control Textiles” addresses an interesting topic and describes clearly the experimental methods and results. Therefore, it may be recommended for publication after minor revision.
A: We thank the reviewer for the positive feedback and we addressed the concerns as follows.
- Authors should improve Figure 8: add important peaks and describe functional groups;
A: We modified Figure 8 according to the suggestion:
- Authors should broaden the discussion of the chemical characterization results (lines 368-276);
A: We added the following explanation (with reference) to Figure 8: " we can find that 2950 cm−1 and 2868 cm−1 are the asymmetric and symmetric stretching vibrations of CH3, and 2916 cm−1and 2838 cm−1 represent the asymmetric and symmetric stretching vibrations of CH2, respectively [59]."
- The following publication is recommended in order to fulfill this section: - Materials Science and Engineering: C, 2019, 100: 837-844.
A: We added the reference to Figure 8 as follows: "1106 cm−1 represents the stretching vibration of the ether bond [60]"

Reviewer 3 Report
- Why the authors choose only PEG 1000 not 1500, 2000, 3000, 4000, 6000
- Is the phase change enthalpies and temperatures of PP improve by increasing PEG molecular weights?
- Authors should mention the Porosity and Pore Size Distribution
- Authors should compare their results with other reported organic, inorganic and carbonaceous materials based PCM
- Authors should include more information about thermal conductivity, because in most of the cases it decreases slightly as the temperature rises
- Authors should mention phase change enthalpies
- Authors should calculate the heat transfer resistance
- Is the thermal conductivity of the PCM-filled PP filament samples is high when compared to PP alone?
- Authors should provide more references to compare their results
Author Response
A: We thank the reviewer for the feedback and we addressed the concerns as follows.
- Why the authors choose only PEG 1000 not 1500, 2000, 3000, 4000, 6000
A: There are two reasons for choosing PEG1000: (i) Liquefying PEG for microfluidic injection becomes increasingly challenging for higher molecular weights. (ii) PEG 1000 has a phase change temperature close to the comfort temperature of the human body.
- Is the phase change enthalpies and temperatures of PP improve by increasing PEG molecular weights?
A: In our work, it was not intended to study the effects of different PEG molecular weights, but to propose a new PCM processing method and to study the resulting phase change behavior.
- Authors should mention the Porosity and Pore Size Distribution
A: We guess the reviewer asks about porosity and pore size distribution of the knitted fabric. Since the fabric was produced for the purpose of infrared testing only, these fabric properties are of no concern for this study.
- Authors should compare their results with other reported organic, inorganic and carbonaceous materials based PCM
A: Considering enthalpy, inorganic PCMs have some advantages. However, in our study, we only consider PCMs which can be used in melt-spinning as compared to finishing technologies, because of the process' simplicity and the fiber's robustness and durability. Considering the challenge of melt-spinning, organic PCMs like hexadecanol have too high vapor pressures at the required melt-spinning temperatures and can thus not be used. Also, the high PCM loading we report in our study could not be achieved by alternative PCMs that require techniques like microencapsulation, blending or coating.
- Authors should include more information about thermal conductivity, because in most of the cases it decreases slightly as the temperature rises
A: We tested the thermal conductivity of fabrics with and without PCM, but the results are strongly influenced by the texture of the textile, which is different with or without PCM fibers added. However, thermal conductivity is not a key discussion point of our manuscript.
- Authors should mention phase change enthalpies
A: We have listed the phase change enthalpies in Table 1.
- Authors should calculate the heat transfer resistance
A: We tested the heat transfer resistance, but the result was strongly influence by the texture of the textile (see former remark regarding thermal conductivity), and, again, heat transfer resistance is not a key discussion point of our manuscript.
- Is the thermal conductivity of the PCM-filled PP filament samples is high when compared to PP alone?
A: The thermal conductivity of our PCM-filled PP filament samples is different compared to pure PP filaments. For PP, the thermal conductivity is 0.11–0.16 W(mK)-1 at 296 K and 0.2 W(mK)-1 at 348 K (https://doi.org/10.1016/B0-12-227410-5/00311-2). For PEG, the value is 0.297 W(mK)-1(https://doi.org/10.1016/j.matchemphys.2013.12.036; https://doi.org/10.1021/je201033f) Again, thermal conductivity is not a key discussion point of our manuscript.
- Authors should provide more references to compare their results.
A: In our manuscript, we focus on the new technology of preparing PCM fibers and on the phase change behavior of melt-spun PP fibers containing PEG1000, which can provide information for further studies in this line of work, i.e. for preparing liquid-core phase change fibers, rather than on the comparision of enthalpies of PCMs. The respective research is relatively scarce, with the exception of the following papers: https:// 10.1016/j.ensm.2016.09.006 (existing Reference 5); https://doi.org/10.1007/s10853-018-2722-5 (added as new reference).
We added the sentence: "Compared to other studies, Zhang et al. [5] prepared flexible phase change composite fibers with loading ratios as high as 82%, and Luo et al. [45] prepared flexible paraf-fin/MWCNTs/PP hollow fiber membranes with a maximum paraffin absorption capacity of 52%."
